# How to Differentiate Benign from Malignant Adrenocortical Tumors?

**DOI:** 10.3390/cancers13174383

**Published:** 2021-08-30

**Authors:** Charlotte L. Viëtor, Sara G. Creemers, Folkert J. van Kemenade, Tessa M. van Ginhoven, Leo J. Hofland, Richard A. Feelders

**Affiliations:** 1Department of Surgical Oncology and Gastrointestinal Surgery, Erasmus MC Cancer Institute, 3015GD Rotterdam, The Netherlands; c.vietor@erasmusmc.nl (C.L.V.); t.vanginhoven@erasmusmc.nl (T.M.v.G.); 2Department of Internal Medicine, Division of Endocrinology, Erasmus MC University Medical Center, 3015GD Rotterdam, The Netherlands; s.creemers@erasmusmc.nl (S.G.C.); l.hofland@erasmusmc.nl (L.J.H.); 3Department of Pathology, Erasmus MC University Medical Center, 3015GD Rotterdam, The Netherlands; f.vankemenade@erasmusmc.nl

**Keywords:** adrenal tumors, adrenocortical carcinoma, diagnostics, molecular markers

## Abstract

**Simple Summary:**

Adrenocortical carcinoma is a rare cancer with a poor prognosis. Adrenal tumors are, however, commonly identified in clinical practice. Discrimination between benign and malignant adrenal tumors is of great importance to determine the appropriate treatment and follow-up strategy. This review summarizes the current diagnostic strategies and challenges to distinguish benign from malignant adrenal lesions. We will focus both on radiological and biochemical assessments, enabling diagnosis of the adrenal lesion preoperatively, and on histopathological and a wide variety of molecular assessments that can be done after surgical removal of the adrenal lesion. Furthermore, new non-invasive strategies such as liquid biopsies, in which blood samples are used to study circulating tumor cells, tumor DNA and microRNA, will be addressed in this review.

**Abstract:**

Adrenocortical carcinoma (ACC) is a rare cancer with a poor prognosis. Adrenal incidentalomas are, however, commonly identified in clinical practice. Discrimination between benign and malignant adrenal tumors is of great importance considering the large differences in clinical behavior requiring different strategies. Diagnosis of ACC starts with a thorough physical examination, biochemical evaluation, and imaging. Computed tomography is the first-level imaging modality in adrenal tumors, with tumor size and Hounsfield units being important features for determining malignancy. New developments include the use of urine metabolomics, also enabling discrimination of ACC from adenomas preoperatively. Postoperatively, the Weiss score is used for diagnosis of ACC, consisting of nine histopathological criteria. Due to known limitations as interobserver variability and lack of accuracy in borderline cases, much effort has been put into new tools to diagnose ACC. Novel developments vary from immunohistochemical markers and pathological scores, to markers at the level of DNA, methylome, chromosome, or microRNA. Molecular studies have provided insights into the most promising and most frequent alterations in ACC. The use of liquid biopsies for diagnosis of ACC is studied, although in a small number of patients, requiring further investigation. In this review, current diagnostic modalities and challenges in ACC will be addressed.

## 1. Introduction

Adrenal lesions are commonly encountered in clinical practice and often incidentally discovered. The prevalence of adrenal lesions increases with age, with incidence rates up to 10% at the age of 70 years [1,2,3,4]. The incidence of adrenal incidentalomas is still rising due to the extensive use of imaging studies in daily practice. Differential diagnosis of an adrenal mass includes a wide spectrum in which adrenocortical hyperplasia, adrenocortical adenomas, myelolipoma, pheochromocytoma, adrenocortical carcinomas (ACCs), adrenal metastasis, adrenal bleeding and tuberculosis should be considered [5]. Adrenocortical carcinoma is a rare cancer with an annual incidence estimated between 0.5 and 2 cases per million population [6,7]. From all adrenocortical carcinomas, about 10–25% of cases are diagnosed incidentally. Its prognosis remains very poor, particularly in the case of metastatic disease [8]. The average age at diagnosis of patients with ACC ranges from 47 to 55 years old, whereas some series demonstrate a bimodal age distribution with a second peak in the pediatric group [9,10]. 

Two key considerations in the diagnostic workup of patients with an adrenal mass are the assessment of hormonal activity and the possibility of malignancy. Therefore, every patient with a suspected ACC should undergo careful clinical assessment, detailed biochemical work-up and adrenal-focused imaging prior to surgery [11]. For decades, there has been debate regarding the optimal diagnostic strategy for patients with possible ACC. Several imaging techniques, like computed tomography (CT), magnetic resonance imaging (MRI) and positron emission tomography with 18F-2-deoxy-D-glucose (mostly combined with CT; FDG-PET/CT) are being used for identifying the origin and biological behavior of the adrenal mass and ultimately guide the decision on adrenalectomy. 

In cases with metastases, either regional or distant, the diagnosis of malignant adrenocortical tumors is evident. However, in cases with only local disease, the Weiss score, consisting of several histopathological criteria, is still the gold standard to separate benign from malignant adrenocortical tumors [11,12]. The European Network for the Study of Adrenal Tumors (ENSAT) classification (Table 1) is the currently used staging system for ACC [13] and has an accuracy of 83% in predicting 3-year cancer-specific mortality [8]. Advanced ACC is defined by locoregional spread (stage III) or distant metastases (stage IV) and represents 18–26% and 21–46% of ACC patients at diagnosis, respectively. 

Several molecular targets of interest have been identified by genome wide sequencing studies providing alternative ways to diagnose adrenocortical carcinoma. Early and correct classification is relevant to establish the appropriate therapeutic strategy and duration and intensity of follow-up. In recent years, as a result of extensive research and international collaborations, existing diagnostic tools for ACC have been improved, and new approaches have been proposed, which will be addressed in this review. 

## 2. Imaging Strategies to Differentiate Benign from Malignant Adrenocortical Tumors 

Currently, three imaging modalities are available for differentiating benign from malignant adrenocortical tumors: CT, MRI and FDG-PET/CT. Once an adrenal mass is discovered, CT represents the first-level imaging modality. In case of high suspicion of an ACC, a chest CT should be performed as well to assess the presence of any pulmonary metastases, because it can guide clinical decisions [11]. 

Unenhanced CT can assess the lipid content in the adrenal mass, which serves as the basis in assessing malignancy by imaging. A Hounsfield unit (HU), the attenuation value at unenhanced CT, equal to or less than 10 is specific for lipid rich lesions and therefore has a high specificity for adenomas. This cutoff was based on a systematic review based on limited data by Dinnet et al. [14]. However, there is still lack of accuracy since lipid-poor and hemorrhagic adenomas can be misdiagnosed [15]. From a recent study in 2017 that prospectively recruited patients with an adrenal mass, it appeared that a tumor diameter of 4 cm and unenhanced CT tumor attenuation of 20 HU are the most appropriate threshold values for consideration of an ACC (Figure 1) [3]. Sensitivity remained the same, but the specificity improved by increasing the cutoff from 10 to 20 HU. This large study confirms data from a study in a French cohort in 2018, which showed a positive predictive value (PPV) of 98.6% for diagnosing adrenal adenoma using these two criteria [16]. These adjusted cutoff criteria would ultimately help to prevent unnecessary imaging or surgeries. The most recent ESE and ENSAT guidelines still state that only in case of homogeneous adrenal lesions with HU < 10 an ACC can be sufficiently reliable ruled out [11], but guidelines are planned to be adjusted based on these recent findings. 

Oncocytic adrenal neoplasms are rare tumors, in which imaging techniques have not been extensively studied. Several factors often used to discriminate adrenocortical carcinomas from adenomas can be used in oncocytic adrenal lesions as well, like homogeneity, size, and percentage enhancement washout characteristics. In contrast, the lipid content cannot be used to distinguish benign from malignant oncocytic tumors [17]. Larger numbers of cases are needed to confirm the conclusions and better understand these adrenocortical tumors. 

Besides characteristics on unenhanced CT, studies have shown that adrenal adenomas have high washout rates on contrast-enhanced CT, making this the next diagnostic step in case of lipid-poor adrenal masses [18,19,20]. Additionally, compared to metastases and pheochromocytomas, adenomas have faster washout rates [15,21]. Cut-off values of an absolute washout >60% and a relative washout >40% have been proposed to accurately diagnose adenomas [19].

Several other radiological criteria, like appearance (i.e., integrity and invasiveness), heterogeneity, the borders, and size of the lesion, are being used. However, the true clinical relevance and clear cutoff values are not yet determined due to large differences between studies and methodological concerns. Necrosis, calcifications, hemorrhage foci and heterogeneity are suggestive of a malignant tumor. Size is thought to be one of the most important predictive factors for malignancy. For a tumor of 4 cm, the specificity and sensitivity to diagnose adrenocortical carcinoma is 52% and 96%, respectively. For an adrenal mass of 6 cm, specificity and sensitivity increase up to 80% and 90% respectively [22]. 

If the characteristics on unenhanced and contrast-enhanced CT do not show a classical ACC appearance, MRI can provide additional information in the assessment of malignancy [23]. MRI can be used in particular for precise localization of the tumor and for assessment of separation from the surrounding structures. Furthermore, a small group of adenomas with minimal lipid content and therefore a HU value >10 can be distinguished with chemical shift sequences on MRI [23]. FDG-PET/CT was first evaluated in patients with known extra-adrenal malignancies. In a systematic review investigating diagnostic value of imaging techniques in adrenal masses, the performance of 18F-FDG PET-CT or MRI for diagnosing malignancy was not better than CT [14]. A retrospective study in 106 patients showed that only for a minority (~5%) of patients undergoing FDG-PET/CT for evaluation of an adrenal mass, the scan would have changed the clinical management [24]. Several studies have, however, reported the use of FDG-PET/CT for assessing malignant potential, although it does not provide robust information on the origin of the adrenal mass (e.g., metastases, pheochromocytoma, primary adrenal) [25,26,27]. This imaging modality should therefore be considered as an additional imaging technique and is primarily useful in the case of suspected malignant disease [28]. With a sensitivity of 91% and a specificity of 97% for the assessment of malignancy of adrenal lesions, FDG-PET is thought to be especially helpful to rule out any diagnosis of ACC or metastatic disease [29]. In patients with a history of an extra-adrenal malignancy, the diagnosis of suspected metastases should only be made after careful consideration of other causes.

Despite a significant number of adrenal incidentalomas that are not reliably characterized by these standard imaging procedures, the previously mentioned factors together will guide the decision to perform adrenalectomy in the individual patient. In the case of undetermined diagnosis by imaging techniques, these imaging factors may be combined with hormonal assessment as also stated in the following paragraph, rapid clinical deterioration or young age, which can point towards ACC. 

In cases in which surgery is not directly performed but the benign nature cannot be established with certainty, the current recommendation is to repeat imaging after 6–12 months [30]. Lack of growth at follow-up makes a malignant mass highly unlikely. We do have to acknowledge that in a retrospective analysis of tumors preceding the diagnosis of ACC, the growth pattern appeared to be highly variable between patients, with long-term stability of size up to 8 years in single cases [31]. However, these cases are considered exceptional. 

In an attempt to find new modalities to discriminate ACA from ACC, molecular imaging has been developed with radiotracers that selectively bind to adrenocortical tissue, like metomidate [32]. Metomidate (MTO) is an imidazole-based methyl ester derivate of etomidate and is a potent inhibitor of CYP11B1 and CYP11B2. These enzymes are highly expressed in the zona fasciculata and glomerulosa of the adrenal cortex. Several studies have shown that MTO-PET particularly differentiated adrenal from non-adrenal origin of the tumor, with a sensitivity of around 90% and a specificity of 96% [33,34,35]. 

[^123^I]-Iodometomidate ([^123^I]IMTO) has been developed as tracer for single-photon emission computed tomography (SPECT) [36]. Two prospective studies have shown that this imaging modality could identify adrenocortical origin of local or metastatic adrenal tumors with a sensitivity of 38–83% and a specificity of 86–100% [37,38]. Especially a large percentage of metastases fails to show [^123^I]IMTO uptake. Due to tumor necrosis or de-differentiation resulting in reduced expression of CYP11B enzymes, either in progressive tumors or in response to medical treatment, a substantial percentage of tumors fails to detect [^123^I]IMTO and it seems to have a limited value in detecting small lesions. In addition, [^123^I]IMTO SPECT is not able to differentiate metabolically active benign and malignancy primary adrenocortical lesions, but the high specificity makes it a potential candidate for more widespread clinical application. The specific uptake of these tracers in tissues from adrenocortical origin has not only been held for diagnostic utility, but also potentially provides a method of selection of patients who may respond to ^123^I-iodometomidate targeted radionuclide therapy. 

## 3. Hormonal Evaluation

### 3.1. Biochemical Diagnostic Procedures

For all patients with (suspected) ACC, a thorough and complete hormonal evaluation is recommended before surgery, regardless of the clinical phenotype. Adrenocortical tumors can produce several hormones, like mineralocorticoids, glucocorticoids, sex steroids, and adrenocortical steroid hormone precursors. Patients should undergo a careful clinical assessment for symptoms and signs of adrenal hormone excess. Additionally, for all patients with suspected ACC, a thorough and complete hormonal evaluation is recommended before surgery, regardless of the clinical phenotype [39]. There are multiple reasons which make a biochemical evaluation before surgery necessary [40]. Firstly, in case of severe complicated hypercortisolism, patients are preoperatively treated with cortisol-lowering therapy, and these patients may require hydrocortisone replacement therapy post-surgery. Furthermore, the steroid hormone profile can assist in the assessment of the risk of malignancy, as described below. Furthermore, elevated hormone concentrations can be used as a biomarker after therapy [41]. Lastly, it is important to rule out a pheochromocytoma before surgery to prevent complications during surgery. 

Primary aldosteronism is rare in ACC and is usually accompanied by severe hypokalemia, whereas the production of androgens, resulting in virilization, and estrogen evidently point towards malignancy in adrenocortical tumors [42]. In approximately half of the patients, the first symptoms of ACC are related to hormonal overproduction, primarily hypercortisolism. In other cases, patients may present with symptoms or complaints as a result of local tumor growth or metastases, like abdominal discomfort, back pain or abdominal fullness [43]. 

### 3.2. Urine Steroid Metabolomics

In recent years, research has focused on discriminating ACC from ACA using urine steroid metabolomics and especially steroid precursors by mass spectrometry coupled with chromatography. The rationale behind is potential dedifferentiation of adrenocortical carcinomas with changes in functionality of adrenocortical enzymes. Consequently, complete steroidogenesis is lost, resulting in a secretory biological signature with accumulation of precursors. Several small reports have indeed shown increased concentrations of steroid precursors produced by ACC compared to benign adrenocortical tumors [44,45,46]. In 2011, a retrospective proof of concept study in 45 ACC patients and 102 ACA patients showed that urine steroid metabolomics could identify 90% of ACC using gas chromatography mass spectrometry (GC-MS) [47]. The study defined 11-deoxycotisol metabolite tetrahydro-11-doxycortisol (THS) as the most discriminative marker. In this study, however, also metastatic ACC cases were included, which do not represent real diagnostic challenges in clinical practice. The findings of this study were confirmed by several other studies [48,49]. It is important to emphasize that there were no differences between functioning and non-functioning ACA. In the recent large multicenter prospective EURINE-ACT trial, the value of urine metabolomics for discriminating benign from malignant adrenocortical tumors was confirmed in 2017 patients [3]. In this study, the panel of urinary steroids, measured by liquid chromatography with tandem mass spectrometry (LC-MS/MS), was analyzed by a machine learning-based algorithm. The combination of the urine steroid metabolomics including several steroid metabolites with tumor diameter and imaging characteristics resulted in the highest predictive value for malignancy compared to either of the factors alone [3]. The PPV was 76.4% and negative predictive value 99.7% for ACC. The main challenge is the difficult implementation of mass spectrometry in clinical practice and the cost-effectiveness. Thereby, patients using drugs altering steroid synthesis or metabolism were excluded, limiting the applicability in this group of patients. However, this multiparametric approach provides an interesting and promising new tool for the decision to perform adrenalectomy, but the real clinical benefit remains to be demonstrated. 

## 4. Histopathology

Despite the intensive radiological and biochemical workup of adrenal lesions, histopathology remains the gold standard for diagnosing ACC [11]. Histopathological evaluation is generally performed on resected adrenal tissues, as biopsies in order to obtain histopathological diagnosis of adrenal incidentaloma are not recommended, except when a patient is inoperable or when an adrenal metastasis is suspected [11]. Adrenal biopsies yield non-diagnostic samples in up to 14% of the patients and adverse events are not infrequent, ranging from pain and discomfort after the procedure to severe life-threatening complications in case of a catecholamine crisis in patients with pheochromocytoma [50,51]. In recent studies, however, adrenal biopsies have gained interest in the diagnosis of lesions with a high suspicion of adrenal malignancy, for example because of suspicion of metastatic disease from a known malignancy elsewhere, with sensitivity and negative predictive value rates ranging from 86% to 100% and 70–100% [51,52,53,54].

The first step in examining a specimen suspect for ACC is determination of adrenal cortical origin, especially in case of a non-functioning tumor, as other tumor types such as renal cell carcinoma, pheochromocytoma and sarcomas may resemble primary adrenocortical tumors and misdiagnosis is not infrequent [55]. The most specific immunohistochemical marker to determine adrenocortical origin is steroidogenic factor-1 (SF-1) with a sensitivity of 98% and specificity of 100% [56,57]. Other markers that can be used for determination of adrenocortical origin are melan-A, inhibin-α, synaptophysin, calretinine and CD56 [56,58]. ACCs are generally negative for common epithelial markers such as cytokeratins and EMA. Furthermore, they are usually negative for CEA and chromogranin [59,60]. The immunohistochemical profile of oncocytic, myxoid and sarcomatoid variants of ACC largely resembles the profile of conventional ACC. Oncocytic ACC are generally synaptophysin positive, half of the tumors are inhibin-α positive and up to one third are melan-A positive [61]. In myxoid ACC, additional neurofilament and CD56 expression can be seen [62]. ACCs with sarcomatoid components may be negative for SF-1 [63]. Furthermore, the spindle cell component is usually negative for melan-A, inhibin-α expression is lower and cytokeratin expression can be observed.

Histopathological discrimination between ACC and ACA comprises both macroscopic and microscopic evaluation. ACC usually appears as a large heterogeneous tumor with the presence of fibrous bands. Furthermore, ACC can show signs of hemorrhage, necrosis and calcifications [58]. ACA on the other hand generally has a diameter of <5cm, is more homogenous and well delineated [58]. Multiple scoring systems and algorithms have been developed to separate ACC from ACA, of which the most widely used systems and scores will be discussed here in detail (Table 2) [64,65,66,67,68,69,70,71,72].

### 4.1. Weiss Score 

The Weiss score is currently the most widely used scoring system for ACC and represents the gold standard [12,65,67]. It consists of nine domains, which can all be scored with either zero points or one point (Table 2). An adrenal tumor with a Weiss score of ≥3 is considered malignant, whereas a Weiss score of 0 or 1 is considered benign. There is a grey zone for adrenal tumors with a Weiss score of 2 or 3 that are considered borderline malignant. In these tumors, the Weiss score cannot discriminate between benign and malignant tumors, with misclassification in 9–13% of the cases [55,73]. Other drawbacks of the Weiss score comprise its decreased diagnostic potential in variants of ACC other than conventional ACC, and a large interobserver variability and subjectivity, diminishing the reproducibility of the score. This can be improved through intensive coaching of pathologists assessing the Weiss score, for example using virtual microscopy and doing RING studies with colleagues, increasing the sensitivity of the Weiss score from 86% to 95% [74]. Hence, the current guidelines recommend the Weiss score to be determined by expert pathologists only [11]. 

### 4.2. Revised Weiss Score

In a revised and simplified Weiss score, only five instead of nine items are scored, with a maximum score of seven (Table 2) [68]. The revised Weiss score correlates well with the original score (r = 0.98) and it has proven to be easier for use in clinical practice and more reproducible than the original Weiss score with similar sensitivity (100% vs. 100%) and specificity (96.9% vs. 90.2%), compared to the original Weiss score to diagnose ACC [58,68,72]. 

### 4.3. Lin-Weiss-Bisceglia System

Particularly for the oncocytic subtype of ACC patients, the Lin-Weiss-Bisceglia system is currently used [70]. Criteria in this system have a higher focus on invasiveness and mitotic count (Table 2). Due to the specific composition, oncocytic tumors would score at least 3 on the Weiss score regardless their clinical behavior. Using the Lin-Weiss-Bisceglia system instead of the Weiss score thereby prevents overdiagnosis of ACC in this subtype of ACC [75]. 

### 4.4. Reticulin Algorithm

Besides morphological criteria that are also used in other scores for ACC (Table 2), the reticulin algorithm uses a histochemical stain to assess the reticulin network, as this is commonly disrupted in ACC [71]. Generally, the quantitative assessment of loss of reticulin network is thought to be more valuable, as underestimation of malignancy can occur when reviewing qualitative changes in reticulin structure [56]. Namely, the extent of reticulin network disruption is heterogeneous, therefore multiple samples should be assessed to prevent missing any signs of reticulin network disruption [76]. It is essential to use the assessment of reticulin network in combination with other histopathological markers, since disruption on the reticulin network has also been described in ACA. In this algorithm, a tumor should exhibit at least one of the other parameters, consisting of >5 mitoses/50 high power fields, presence of necrosis and venous invasion. In the original study that proposed this algorithm for diagnosing ACC, 84% of the tumors that were eventually considered as ACC even exhibited two of the parameters next to the disrupted reticulin network [71]. The algorithm is validated in adults with conventional ACC with a sensitivity and specificity of 100% and an interobserver agreement rate of 75%, which even increased after training to 86% [71,77]. Moreover, validation was undertaken for oncocytic ACC [76,78,79]. Advantages of the reticulin algorithm comprise the objectivity of this scoring system and the low number of parameters needed to determine the risk of malignancy. 

### 4.5. Helsinki Score

A rather recently developed score, the Helsinki score, focuses on a combination of the Ki67-index, which will be discussed in more detail later, the mitotic rate and the presence of necrosis for diagnosing ACC (Table 2) [72]. The cut-off for malignancy was originally set at >8.5 points, resulting in a sensitivity of 100% and a specificity of 99.4% [72]. Later, the Helsinki score was validated (AUC = 0.729, compared to an AUC = 0.624 for the Weiss score) and new cut-off values of <13 and >19 were proposed to optimize the prognostic stratification value of the score [80]. The Helsinki score can be used not only for diagnosing conventional ACC, but also for oncocytic and myxoid variants [80]. Although in a French cohort the diagnostic performance of the Helsinki score was less accurate for diagnosis of the oncocytic subtype, it could be used for defining prognostic subgroups within the group of oncocytic ACC patients [76].

### 4.6. Ki67-Index

Ki67 is a proliferation marker that can be assessed with immunohistochemistry using the MIB-1 antibody. As proliferation is a common feature in malignant tumors, Ki67 is usually overexpressed in tumor tissue. The diagnostic and prognostic potential for ACC of the Ki67-index has been shown in multiple studies [10,60,81,82,83,84,85,86]. Therefore, the most recent guidelines recommend to assess the Ki67-index in every resection specimen of adrenal cortical tumors [11]. A MIB-1 labeling index of 4% yielded a sensitivity of 95.7% and a specificity of 91.7% for diagnosing ACC [68]. In general, the cut-off for assessing malignancy based on the Ki67-index is set at >5% [87], which leads to appropriate separation of adenomas and carcinomas [83,88,89]. As the proliferation rate is generally heterogeneous across a tumor, the Ki67-index should be assessed in the region with the highest proliferation rate (so-called “hot spots”). When selecting this region visually, interobserver concordance is limited [90]. Multiple techniques, such as automated selection of hotspots (ASH), are developed to objectify and thereby optimize region selection for Ki67-index measurements [90,91]. 

### 4.7. Other Immunohistochemical Assessments

Immunohistochemistry is of increasing importance as tumors might be fragmented or morcellated after being removed during laparoscopic adrenalectomy which hampers the assessment of invasive growth because of the proliferative heterogeneity of these tumors. An important general remark, however, on the diagnostic value of immunohistochemical staining, is potential interlaboratory differences which may occur due to different staining protocols and assessment methods [72].

Among markers that can distinguish ACC from benign adrenal neoplasms, insulin-like growth factor 2 (IGF-2) is the most frequently studied, with overexpression in up to 90% of ACC (see also sections ‘Methylome’, ‘Transcriptome’ and ‘DNA mutations’). Interestingly, a specific juxtanuclear Golgipattern of IGF-2 expression is seen in ACC, which is most likely caused by impairment of translation and processing of IGF-2 molecules within the Golgi apparatus, leading to decreased secretion of mature IGF-2 and increased secretion of heavier precursor forms [56,58,84,92,93]. This was shown to be discriminant between ACC and benign lesions with a sensitivity of 76.5% and specificity of 95.5% [84]. Additionally, the combination of IGF-2 with other markers, such as Ki67, MAD2L1 or CNNB1 expression can be used to diagnose ACC, as these combinations provide a sensitivity of 96%, 100% and 91% and a specificity of 100%, 95% and 100% for diagnosing ACC, respectively [60]. In another study, sensitivity and specificity of IGF-2 with MIB-1 staining (which is generally used to assess Ki67 expression) were 100% and 95.5%, respectively [84]. MIB-1 staining alone yielded a sensitivity of 87.5% and a specificity of 95.5% for discriminating between ACC and ACA in this study [84]. 

Genomic analyses, which have demonstrated several other genes with differential expression in ACC and ACA (see also section ‘Transcriptome’ and ‘DNA mutations’), are expensive and not always easily accessible. Therefore, based on these findings, studies have focused on immunohistochemical analyses of markers that have shown increased or decreased expression in ACC. Higher expression of markers for cell proliferation and mitotic spindle regulation (Ki67, p53, BUB1B, HURP and NEK2) has been described, as well as higher expression of DNA damage repair markers (PBK and y-H2Ax) and global loss of expression of regulators of telomeres (DAXX, ATRX) [56]. P53 was predominantly found to be specific for diagnosing ACC (specificity 100%), but not sensitive (17.6%), as either overexpression or global loss of TP53 expression is only seen in 20–25% of ACCs [10,58,84,94]. Furthermore, a differential diffuse nuclear or cytoplasmic accumulation of β-catenin is a common feature of ACC (see also section ‘DNA mutations’) [56,95]. Lastly, phosphohistone-H3 (PPH3) immunostaining is an accurate and fast way to determine mitotic index and to assess atypical mitoses, features that are incorporated in almost all ACC diagnostic scoring systems [86]. 

## 5. Diagnostic Molecular Biomarkers

Over the last decades, possibilities for assessment of the molecular characteristics of tumors have increased and multiple studies have focused on characterizing the genomic aspects of ACC. ACC is found to be a heterogenous form of cancer with a large genetic diversity. This is reflected by results of analysis of the methylome, transcriptome, miRNAome and alterations in the DNA itself, as described below. As a result of the large variability regarding these molecular markers across ACC tumors, the use of molecular biomarkers to diagnose ACC has not yet been incorporated in clinical practice. With the expanding possibilities for analysis of such markers, however, it has become a very promising subject of research that might aid clinical practice in the near future.

### 5.1. Methylome 

Methylation of certain regions within the DNA is one of the major regulators of gene expression. Genomic instability, loss of parental imprinting and reactivation of transposable elements due to global hypomethylation have been observed in many cancer types. Moreover, hypermethylation of CpG-islands, which are DNA sequences with a large number of cytosine associated with guanine repeats usually found within the promoter of genes, is seen as well in cancer, resulting in transcriptional inactivation of tumor suppressor genes [96]. 

Genome-wide studies comparing methylation in ACC and benign adrenal tissues show both global hypomethylation and hypermethylation of CpG-islands of genes involved in apoptosis regulation, transcription regulation and cell cycle control, such as *CDKN2A*, *GATA4*, *BCL2*, *DLEC1*, *HDAC10*, *PYCARD*, *SCGB2A1/HIN1*, *KCT12*, *KIRREL*, *SYNGR1*, *NTNG2* and the imprinting region with *IGF2* and *H19* on chromosome 11p15 [97,98,99].

Studies on methylation of targeted candidate genes in the oncogenesis of ACC highlighted predominantly an increased methylation of the *H19* region, which is located next to the *IGF2* gene on chromosome 11, leading to *IGF2* overexpression [100]. However, *H19* methylation and *IGF2* expression alone could not discriminate between ACC and ACA in all cases, with a sensitivity and specificity around 80% [101]. Therefore, Creemers et al. proposed an *IGF2* methylation score, not only based on *H19* methylation, but also on methylation of two other main regions (*CTCF3* and *DMR2*) within the *IGF2* promoter [101]. This score proved to be discriminative for ACC with an AUC of 0.910 (95% CI 0.866–0.952) and was later validated in a large European cohort (Figure 2) [102]. Combined together with tumor size, the methylation score even yielded an AUC of 0.957 (95% CI 0.930–0.984) for discriminating between ACC and ACA (Figure 2) [102].

### 5.2. Transcriptome 

Understanding of differences in gene expression profiles in ACC compared to benign adrenal lesions can be used to confirm pathological diagnosis of ACC. Furthermore, it can be valuable in hospitals with limited experience in histopathological examination of ACC due to the rarity of the disease, and it might provide additional diagnostic information that cannot be obtained through histopathological assessment alone. Therefore, interest in understanding transcriptomic deregulation in ACC has grown over the years. Many studies have used unsupervised clustering to assess differences between ACC and ACA on the level of gene expression. In general, they found a heterogeneous gene expression profile in ACC that could reliably discriminate ACC from ACA, with predominantly deregulated expression of genes involved in cell cycle regulation, chromosomal maintenance, cell survival, inflammation and immunity in ACC [103,104,105,106,107,108,109]. Giordano et al. found 2875 genes with differential expression in ACC compared to ACA, including *IGF2*, *SPP1*, *TOP2A*, *ENC1*, *H19*, *CNNB2*, *ASDM*, *RRM2* and *CDKN3* [106]. A dysregulated gene expression of at least 1017 of these genes was confirmed by multiple other studies [108,110,111]. Laurell et al. demonstrated differential expression of *IGF2*, *FGFR1* and *FGFR4* as well as genes involved in the ubiquitin protease pathway such as *USP4*, *UBE2C* and *UFD1L* [112]. Furthermore, differential gene expression of *IL13RA2*, *HTR2B*, *CCNB2*, *RARRES2* and *SLC16A9* was discriminative between ACC and ACA with an AUC of the ROC curve of >0.80 [113].

The most consistent finding across studies involves upregulation of *IGF2* expression which has been shown in at least 85% of ACC compared to ACA [114,115], which was confirmed by immunohistochemistry [60,116]. In a study of de Fraipont et al., adrenocortical lesions could be divided into an IGF-cluster (expression of eight IGF-associated genes) and a steroidogenesis cluster (expression of 14 steroidogenesis-related genes). Tumors that showed high within the IGF-cluster were ACC in 75% of the cases, whereas tissues scoring low on the IGF-cluster expression were ACA in 90%. With regard to the steroidogenesis cluster, tumors with low expression proved to be ACC in 81% of the cases, whereas tumors with high steroidogenesis-associated gene expression were ACA in 93% of the cases [108]. 

Expression of other genes has been implied to be discriminative of malignancy as well. De Reynies et al. stated that the combined overexpression of *DLG7* and underexpression of *PINK1*, genes involved in cell cycle regulation, could discriminate between ACC and ACA with at least similar accuracy to the Weiss score. Using this combined expression, 98% of the benign and 96% of the malignant adrenal tumors could be reliably diagnosed. Furthermore, using this combined expression score, 86% of the tumors with a Weiss score of 2/3 were classified as malignant [117]. This gene expression algorithm might therefore aid in the diagnostic workup of suspicious adrenal lesions, as it might overcome some of the uncertainties of using the Weiss score. Significant upregulation of hepatocyte growth factor (*HGF*) and its receptor cMET has been observed as well in ACC compared to ACA [118]. In other cancer types, upregulation of this pathway is associated with stimulated tumor angiogenesis, enhanced cell proliferation, tumor growth and reduced apoptosis. More recently, Yuan et al. used weighted gene co-expression network analysis (WGCNA) to assess a specific gene co-expression network in ACC that could predict prognosis and tumor grade. They identified 12 hub genes (*ANCN*, *ASPM*, *CDCA5*, *CENPF*, *FOXM1*, *KIAA0101*, *MELK*, *NDC80*, *PRC1*, *RACGAP1*, *SPAG5* and *TPX2*) that, besides predicting prognosis and tumor grade, had good distinctive power to determine malignancy [119].

Long non-coding RNAs (lncRNAs) consist of >200 nucleotides, and do not encode proteins but function as decoys, scaffolds and enhancer RNAs and are involved in chromatin remodeling and (post)transcriptional regulation of gene expression [120]. Recently, Buishand et al. demonstrated in a study of 9 ACC, 11 ACA and 5 normal adrenal tissues that ACCs show a distinct profile of lncRNAs compared to ACA and normal adrenal tissues, with 874 lncRNAs being differentially expressed and associated with prognosis [121]. The true diagnostic potential of lncRNAs should be elucidated in future studies.

### 5.3. MiRNAome

Micro-RNAs are small non-coding RNA sequences that have an important role in the posttranscriptional gene expression regulation through cleavage and translation repression [114]. The role of micro-RNAs in pathogenesis, diagnostics and prognostics has been investigated for several cancer types, among which is ACC [122]. Several studies reported microRNAs being differentially expressed between ACC and ACA, of which upregulation of miR-503, miR-210, miR-483-5p and miR-483-3p and downregulation of miR-195 and miR-335 has been reported in at least two studies [111,123,124,125,126,127,128,129,130]. Results have been inconclusive on other micro-RNAs [123,131]. Deregulated expression of several micro-RNAs was not only associated with ACC diagnosis but also with prognosis [114,124,132,133]. The diagnostic accuracy of micro-RNAs is shown to be variable [134]. miR-483-5p seems to be the best predictor of malignancy, with a sensitivity and specificity of up to 87.5% and 94.4%, respectively [135]. Another study showed a sensitivity of 80% and a specificity of 100% with a PPV of 100% and a NPV of 92% of miR-483-5p to diagnose ACC [126]. Additionally, both miR-483-5p and two other micro-RNAs, miR-195 and miR-335, showed good diagnostic accuracy for ACC in the study of Chabre et al. with an AUC ≥0.830 for all three micro-RNAs [130]. Furthermore, the combined expression of six micro-RNAs (miR-503-5p, miR-483-3p, miR-450-5p, miR-210, miR-483-5p and miR-421) was able to predict malignancy with 95% accuracy [136]. Combinations of miR-511 and miR-503 (sensitivity 100%, specificity 93%) [111], miR-511 and miR-184 (sensitivity 100%, specificity 80%) [111] and miR-483-3p with the immunohistochemical marker Smad4 (specificity 92.8%) [116] can possibly distinguish ACC from ACC as well. However, the diagnostic accuracy of these micro-RNAs has only been assessed in small cohorts of ACC tissues, and should therefore be validated in larger cohorts to assess the true diagnostic value of these markers in clinical practice.

### 5.4. Chromosomal Aberrations 

A variety of chromosomal aberrations, such as chromosomal amplifications, gains, losses and loss of heterozygosity (LOH), has been described in patients with ACC. Compared to ACA, multiple studies reported increased numbers of recurrent copy number variations (CNVs) in ACC [137,138,139,140,141,142]. CNVs had a pattern generally resulting in gains on chromosomes 5, 7, 12, 16, 19 and 20, and losses on chromosomes 1, 2, 13, 17 and 22 [137,138]. These chromosomal aberrations cause gene amplification of *TERT* (5p15.33) and *CDK4* (12q4), and homozygous deletions of the *CDKN2A* (9p21.3), *RB1* (13q14) and *ZNRF3* (22q12.1) gene [139,143,144]. LOH has been described in over 90% of the ACC patients [137], with LOH profiles consistent with the copy number alterations [139,143]. Additionally, frequent LOH or allelic imbalance was observed on loci 11q13, 17p13 and 2p16 [145,146,147]. Zheng et al. showed that whole genome doubling is a common feature in ACC, with associated LOH, which is thought to be associated with more aggressive tumors [139]. Summarizing, the chromosomal alterations that can be seen in ACC are diverse and heterogeneous and comprise large chromosomal regions, thereby complicating the identification of candidate genes to diagnose ACC more reliably. Two studies have, however, addressed the diagnostic potential of chromosomal aberrations itself in diagnosing ACC. Barreau et al. showed that alterations in six loci (on chromosomes 5q, 7p, 11p, 13q, 16q and 22q) could predict ACC diagnosis with a sensitivity of 100% and a specificity of 83% in a validation cohort [140]. Moreover, Ronchi et al. stated that an amplification of >60% of chromosome 5 and the combination of >50 large copy number variations with >10 LOH events both had adequate sensitivity (77.3% and 82%, respectively) and specificity (both 100%) to diagnose ACC [137].

### 5.5. DNA Mutations 

The first studies on mutations causing ACC focused on familial syndromes, such as Li-Fraumeni and Beckwith-Wiedemann syndrome, with a high prevalence of ACC [148]. Later, several studies have presented or validated a mutation profile of sporadic ACC. Frequently altered genes in ACC include *TP53* (~16–36%), *ZNRF3* (~19–21%), *CTNNB1* (~16–19%), *CDKN2A* (~11–15%), *TERT* (~6–14%), *PRKAR1A* (~11%), *RB1* (~7%), *MEN1* (~7%), *DAXX* (~6%), *ATRX* (~4%), *MDM2* (~4%) and *CDK4* (~2%) [139,143,149]. Mutations in other genes, such as *ATM*, *KREMEN1*, *MED12*, *JAK3*, *RPL22*, *TERF2* and *NF1* have also been described [105,139,143,144,150,151,152]. 

A higher prevalence of *TP53* mutation is seen in pediatric ACC compared to adult patients with ACC. Pediatric ACC is extremely frequent in Southern Brazil, where 90% of the patients harbor a *TP53* hotspot mutation p.R337H [114]. Outside of that region, *TP53* germline mutations are described in up to 50% of the pediatric ACC patients, with a decrease in prevalence of the *TP53* mutations with increasing age [153]. Although the prevalence of *TP53* mutations in adult cohorts of ACC patients was considerably lower, loss of heterozygosity of the *P53* gene at locus 17p13 is reported in 85% of sporadic ACC [146], implicating that *TP53* is not the only tumor suppressor gene at that specific locus [154].

The *ZNRF3* gene codes for a cell-surface transmembrane E3 ubiquitin ligase, which is a negative feedback regulator in the Wnt signaling pathway. Mutations in this gene have been described in other cancer types as well, such as colorectal cancer and papillary thyroid carcinoma [155,156]. *CTNNB1* mutations also affect the Wnt signaling pathway. However, in a study of 39 ACC patients, mutations in the *CTNNB1* gene were quite common both in adenomas and in carcinomas, hence presence of the mutation was not discriminative for malignancy. The distribution of β-catenin accumulation was however predictive for diagnosis of ACC, as β-catenin was almost exclusively focally distributed in the adenomas, and a more diffuse nuclear and cytoplasmic β-catenin distribution was seen in ACC [56,95].

The rarity and diversity of mutated genes in ACC have hampered the diagnostic potential of individual mutations. Recently, Zheng et al. investigated whether combinations of mutations can aid ACC diagnosis. Six genes of which the mutation rate was significantly higher in ACC than ACA (*ZNRF3*, *PRKAR1A*, *TP53*, *ARMC5*, *RB1* and *APC*) were denominated as high-risk genes. The sum of high-risk gene mutations (SHGM) was >0 in 73% of ACC tissues and >1 in 62.2% of ACC tissues. In ACC smaller than 5 cm, SHGM >0 was observed in 75% and SHGM >1 in 50% of ACC tissues [157]. This indicates that multiple gene mutations taken together might be valuable in diagnosing ACC, especially when the diagnosis cannot exclusively be made based on histopathological and/or clinical grounds (such as small tumor size). 

### 5.6. Liquid Biopsies

Liquid biopsies are a potential novel diagnostic, prognostic and therapeutic monitoring strategy that enables minimal invasive assessment of biomarkers in a blood sample. Its potential has been established already in multiple tumor types and has recently been investigated in ACC as well. In blood, circulating tumor cells (CTCs), micro-RNAs, exosomes and cell-free DNA (cfDNA) among which also circulating tumor DNA (ctDNA) can be analyzed. The profile obtained though analysis of the cfDNA provides different information compared to CTCs, therefore both measurements can be valuable in distinguishing malignant and benign tumors [158,159,160].

Circulating tumor cells in ACC samples have been found in 68% up to all cases [161,162]. Furthermore, the number of CTCs was found to correlate with the presence of miR-483-5p, which was also present in serum of ACC patients [163]. The diagnostic potential of this and other micro-RNAs in serum has been studied by several research groups (Table 3) and has shown to be quite accurate for miR-34a, miR-483-5p, miR-195, miR-139-5p, miR-335, miR-376a and for combinations of miR-210/181b and miR-100/181b [127,130,135,164,165]. 

To date, the presence of mutations in cfDNA within serum samples of ACC patients was assessed in two studies. The first study was a pilot study in six ACC patients, in which mutations in ACC tissue were found in three patients. Similar mutations as found in the primary tumor tissue were demonstrated in one of these three patients in cfDNA, indicating the presence of ctDNA in plasma of this patient [166]. Garinet et al. also demonstrated the presence of ctDNA only in a subset of ACC patients. In this study, at least one mutation could be detected in the tumors of eight patients, but mutations in the cfDNA were only found in two out of these eight patients [167]. These findings indicate that cfDNA mutation detection can vary among ACC patients, which might be associated with tumor burden and prognosis. Larger patient cohorts and factors associated with the presence of mutations in cfDNA should be explored to assess the potential and relevance for use in clinical practice.

## 6. Conclusions

Recently, several developments have been made to improve the diagnosis of adrenocortical carcinoma. This review forms a broad overview of the currently used modalities and markers that are still under investigation to diagnose ACC. A summarizing overview of the currently used and potential markers to diagnose ACC is shown in Figure 3. The diagnostic potential of the features and markers that are discussed in this review is shown in Table 4. Accurate diagnosis of malignancy is crucial in order to guide the decision on adrenalectomy, for prognosis stratification, and to determine the intensity and duration of follow-up. Adrenal tumors with clear characteristics of either benign or malignant disease can usually be diagnosed accurately. A large group of tumors with unexpected clinical behavior that therefore are in the “gray zone” between ACC and benign disease remains however challenging. Therefore, especially to improve diagnosing tumors within this gray zone, it is necessary to combine all known discriminative features and to keep searching for new discriminative factors. Recent multinational collaborative efforts have made important contributions to improve the diagnosis of ACC. As such, a recent large clinical trial provided a new cut-off value for the Hounsfield Unit for malignancy. Furthermore, urine metabolomics have been suggested as a sensitive diagnostic tool for determining malignancy of adrenal tumors, although further research is necessary to validate these findings. These developments are especially very promising as radiological parameters and urine metabolomics can be assessed preoperatively. Therefore, these tools might aid clinicians in the decision on adrenalectomy. Until now, the gold standard to diagnose ACC remains the Weiss score, assessed by expert pathologists. Attempts have been made to reduce the inter-observer variability when applying the Weiss score, and other algorithms or scoring systems are proposed that are easier to use and more objective, such as the revised Weiss score. Potentially, the use of this score instead of the original Weiss score might enable more accurate ACC diagnosis. Multi-omics studies have furthermore defined the landscape of molecular alterations in adrenocortical tumors. ACC are heterogeneous tumors with large genetic diversity with variances in the methylome, transcriptome, miRNome, and alterations in the chromosomes and DNA. These large studies in general necessitate further validation of specific alterations in order to extrapolate these data to the individual patient, which has been done for some of the most promising markers. To date, however, external validation remains one of the most important challenges of the proposed markers, due to the rarity of ACC. To be able to validate proposed markers for the diagnosis of ACC, large multicenter collaborative initiatives are needed to generate adequate sample sizes to ultimately adjust clinical guidelines. Furthermore, using uniform outcome measures would enable us to adequately compare the performance of different diagnostic markers. Studies that focus on the most challenging diagnostic cases, i.e., adrenal tumors with an uncertain clinical behavior based on the Weiss score, are also urgently needed. In these cases, the challenge lies especially in the lack of a gold standard. Instead, clinical behavior could be used as a surrogate for diagnosis. 

The determination of genetic alterations in body fluids are now also subject of investigation in an attempt to non-invasively determine the biology of adrenal tumors pre-operatively. Further research is necessary to evaluate the value of this method for prognosis or diagnosis of ACC. 

As a result of all mentioned developments and novel insights, guidelines are continuously adapted according to the newest data ultimately in order to improve patient care.

## Figures and Tables

**Figure 1 cancers-13-04383-f001:**
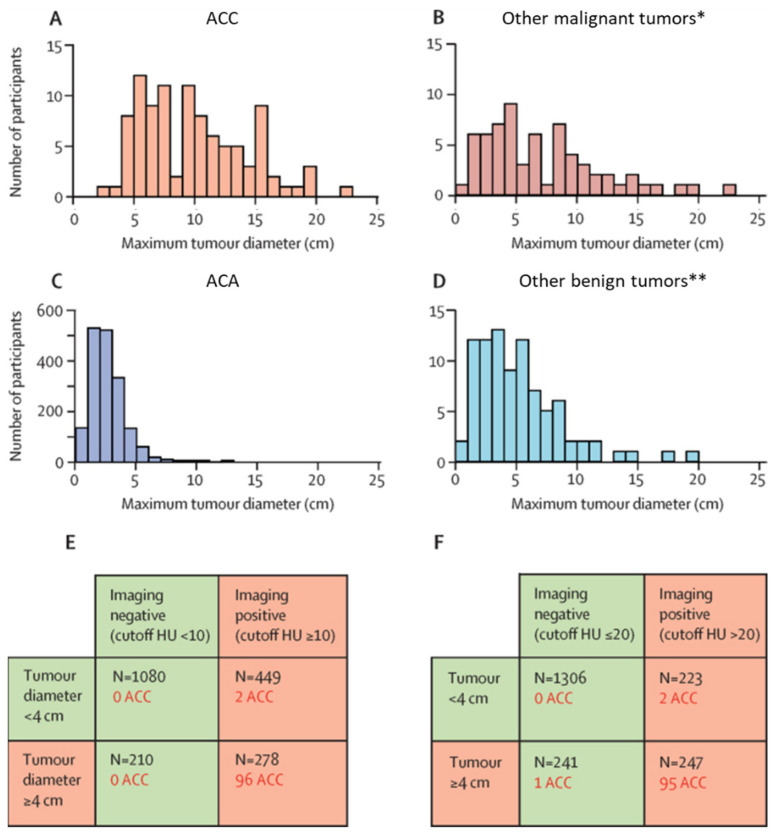
Discrimination between ACC and ACA based on CT imaging characteristics; tumor size and HU attenuation (adapted from Bancos et al. [3]). Distribution of the maximum tumor diameter in patients with ACC (*n* = 98) (**A**), other malignant tumors * (*n* = 65) (**B**), ACA (*n* = 1767) (**C**), and other benign tumors ** (*n* = 87) (**D**) and distributions of patients with ACC according to positive or negative results for tumor diameter and imaging characteristics with unenhanced CT tumor attenuation cutoff of 10 HU (**E**) or 20 HU (**F**). In subfigure E and F, the total number of patients N comprises all patients with ACC, other malignant tumors, ACA and other benign tumors, of which the number of ACC patients is depicted in red. Of the group of adenomas, HU tumor attenuation was measured in 1328 patients. When focusing on discrimination between ACC and ACA, a cutoff HU ≥ 10 resulted in a false positive result in 423 patients (31.9%). A cutoff HU ≥ 20 yielded 200 false positive ACA patients (15.1%). Increasing the cutoff value from HU ≥ 10 to HU ≥ 20 leads to an increase in specificity for diagnosing ACC from 64% to 80%, therefore a cutoff HU of 20 is the most appropriate threshold for malignancy of adrenal tumors. These specific numbers cannot be extracted from the figure, since the total amount of patients also includes other entities of adrenal lesions. ACC = adrenocortical carcinoma, ACA = adrenocortical adenoma, HU = Hounsfield units. * including metastasis, primary adrenal lymphoma, leiomyosarcoma, angiosarcoma, liposarcoma, neuroblastoma, sarcoma, castleman. ** including myelolipoma, cyst, pheochromocytoma, ganglioneuroma, hemangioma, hematoma, schwannoma, lymphangioma, hepatic adenoma, pseudocyst, stromal tumor, angiolipoma.

**Figure 2 cancers-13-04383-f002:**
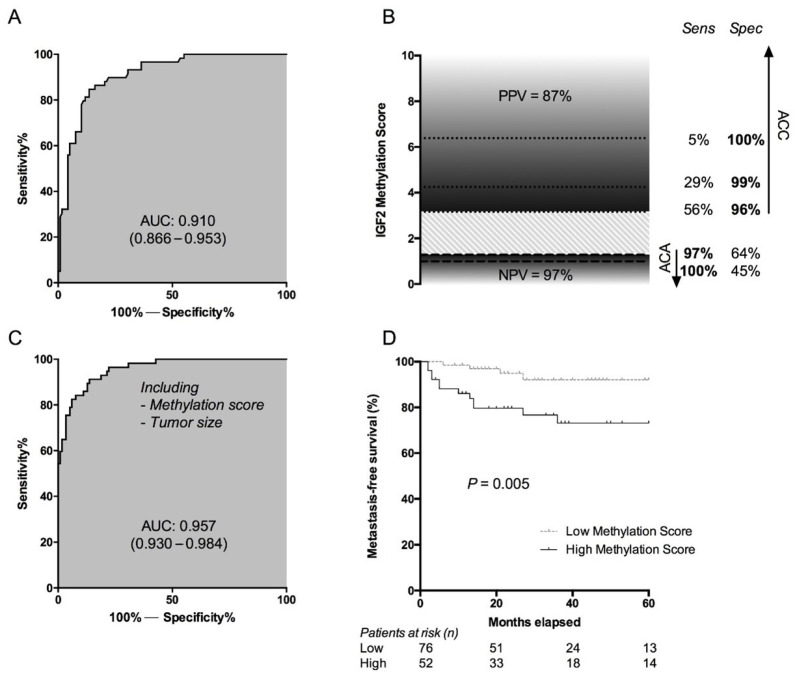
Discriminative value of the IGF2 methylation score for discrimination between ACC (*n* = 59) and ACA (*n* = 118) (cited from Creemers et al. [102]). ENSAT tumor stage IV patients were excluded from analyses (*n* = 17). (**A**) ROC curve of the IGF2 methylation score for prediction of the pathological diagnosis of ACC. (**B**) Sensitivity and specificity for specific cutoff values of the IGF2 methylation score for the pathological diagnosis of ACC. The striped area represents a grey zone of the methylation score with less diagnostic accuracy. PPV and NPV for the cutoff value below (1.28) or above (3.18) the grey zone. (**C**) ROC curve of the logistic regression model including the methylation score and tumor size for predicting the pathological diagnosis of ACC. (**D**) Kaplan-Meier curve for two groups based on the IGF2 methylation score for development of metastases. The two groups were divided based on an IGF2 methylation score of 2.45, which was based on the best discriminative value for the development of metastases calculated using ROC analysis. ACA = adrenocortical adenoma, ACC = adrenocortical carcinoma, AUC = area under the curve, NPV = negative predictive value, PPV = positive predictive value, ROC-curve = receiver operating characteristic curve.

**Figure 3 cancers-13-04383-f003:**
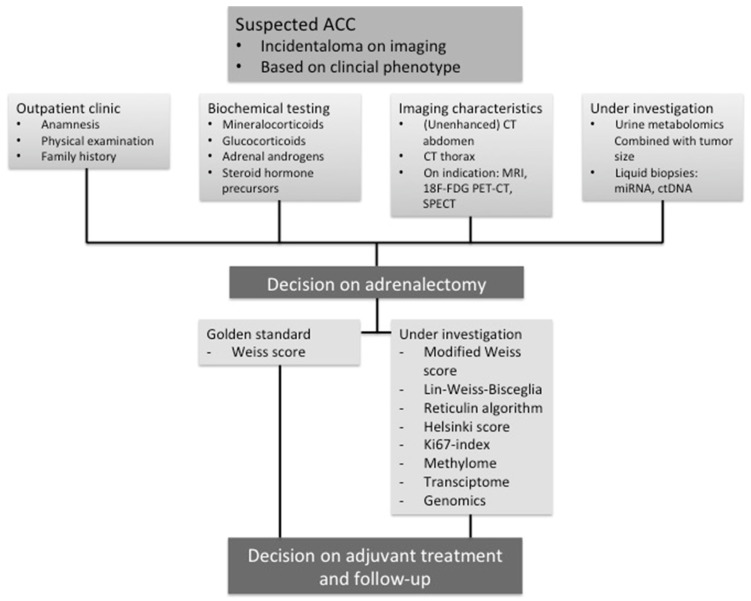
Overview of currently used and potential diagnostic modalities in clinical decision making in patients with suspected adrenocortical carcinoma (ACC). CT = computed tomography, ctDNA = circulating tumor DNA, miRNA = micro-RNAs, MRI = magnetic resonance imaging, SPECT = single-photon emission computed tomography.

**Table 1 cancers-13-04383-t001:** Staging system for ACC from the European Network for the Study of Adrenal Tumors (ENSAT).

ENSAT Stage	T	N	M
I	1	0	0
II	2	0	0
III	1, 2	1	0
	3, 4	0, 1	0
IV	1–4	0, 1	1

Tumors are classified as follows: T1, tumor < 5 cm; T2, tumor > 5 cm; T3, tumor infiltration into surrounding (fat) tissue; T4, tumor invasion into adjacent organs or venous tumor thrombus in vena cava or renal vein; N0, no spread into nearby lymph nodes; N1, positive lymph node(s); M0, no distant metastasis; M1, presence of distant metastasis.

**Table 2 cancers-13-04383-t002:** Different histopathological scores and algorithms for diagnosing ACC.

Parameter	Weiss (1989) [67]	Modified Weiss (2002) [68]	Lin-Weiss-Bisceglia (2004) [70]	Reticulin Algorithm (2009) [71]	Helsinki Score (2015) [72]
**Tumor type(s):**	Conventional	Conventional	Oncocytic	Conventional Oncocytic Myxoid	Conventional Oncocytic Myxoid
**Criteria:**					
Mitosis	>5/50HPF	>5/50HPF	**>5/50HPF**	>5/50HPF	>5/50HPF (x3)
Atypical mitosis	x	x	x		
Necrosis	x	x	**x**	x	x (x5)
Clear cells ≤25%	x	x (x2)			
Venous invasion	x		x	x	
Capsular invasion	x	x	**x**		
Sinusoidal invasion	x		**x**		
Diffuse architecture >30%	x				
FNG 3 or 4	x				
>10 cm and/or >200 g			**x**		
Altered reticulin network				x	
Numeric value of Ki67%					x
**Cutoff for malignancy:**	≥3 points	≥3 points	1 major *	Reticulin + 1	>8.5 points

HPF = high power fields, FNG = Fuhrman nuclear grade; * Major criteria are depicted in bold, in case of 1–4 minor criteria the tumor is of uncertain malignant potential.

**Table 3 cancers-13-04383-t003:** Diagnostic potential of serum micro-RNAs for distinguishing ACC from ACA.

Micro-RNA	Number of Patients	AUC	Sensitivity (%)	Specificity (%)	Reference
miR-34a	17 ACC, 22 ACA	0.81	-	-	[127]
miR-483-5p	17 ACC, 22 ACA	0.74	-	-	[127]
miR-195	14 aACC, 9 naACC, 14 ACA	0.948	90.9	100	[130]
miR-139-5p	14 aACC, 9 naACC, 14 ACA	0.714	87.5	65	[130]
miR-335	14 aACC, 9 naACC, 14 ACA	0.837	95.2	71.4	[130]
miR-376a	14 aACC, 9 naACC, 14 ACA	0.811	71.4	85.7	[130]
miR-483-5p	14 aACC, 9 naACC, 14 ACA	0.929 *	85.7 *	100 *	[130]
miR-210/181b	9 ACC, 8 ACA	0.87	88	75	[164]
miR-100/181b	9 ACC, 8 ACA	0.85	77.8	100	[164]
miR-483-5p	16 ACC, 18 ACA	0.965	87.5	94.4	[135]
miR-483-5p	23 ACC, 23 ACA	0.88	87	78.3	[165]

miR = micro-RNAs, aACC = aggressive ACC, naACC = non-aggressive ACC; * AUC, sensitivity and specificity for discriminating between aACC and naACC.

**Table 4 cancers-13-04383-t004:** Overview of the diagnostic potential of markers to discriminate between ACC and ACA.

Marker	Reference	Number of Patients	Sensitivity (%)	Specificity (%)	Other Accuracy Measures
Radiological Markers					
Diameter ≤4 cm + HU ≤20 ^1^	[16]	233	76.4	96.9	PPV 98.6%, NPV 59.0%
Diameter ≥4 cm	[22]	504	96	52	LR malignancy 4.4
Diameter ≥6 cm	[22]	504	90	80	LR malignancy 16.9
Diameter ≥6.5 cm	[68]	49	100	91.7	
HU ≥ 10	[3]	2017	100	64.0	-
HU ≥ 20	[3]	2017	100	80.0	-
FDG-PET	[29]	1217	97 ^2^	91 ^2^	PLR 11.1, NLR 0.04, OR 294 ^2^
MTO-PET	[35]	173	89 ^3^	96 ^3^	-
I123IMTO	[37,38]	51–58	38–83 ^3^	86–100 ^3^	-
Biochemical Markers					
Urine steroid metabolomics	[47]	147	90	90	AUC 0.97
Imaging + urine	[3]	2017	-	-	PPV 76.4%, NPV 99.7%
Histopathological and Immunohistochemical Markers					
Weiss	[72,74]	50–177	86–100	90.2	Misclassification in 9–13%, AUC 0.624
Revised Weiss	[68,72]	49–177	100	96.9	-
Reticulin algorithm	[71,77]	139–245	97–100	100	-
Helsinki score	[72]	177	100 ^4^	99.4 ^4^	AUC 0.729 ^4^
Ki67/MIB-1 labeling index	[60,68,81,84,116]	37–64	64–100	91.7–100	-
IGF2 expression	[60,84,116]	39–64	64–76.5	72–100	-
IGF2 + Ki67	[60]	34	96	100	-
IGF2 + MAD2L1	[60]	34	100	95	-
IGF2 + CNNb1	[60]	34	91	100	-
IGF2 + MIB-1	[84]	39	100	95.5	-
P53	[84]	39	17.6	100	-
Molecular Markers					
IGF2 methylation score	[101,102]	22–194	89–96	92–100	AUC 0.910–0.997
IGF2 methylation score + tumor size	[102]	194	-	-	AUC 0.957
miR-483-5p	[126,130,135]	31–34	80–87.5	94.4–100	PPV 100%, NPV 92%, AUC 0.90–0.96
miR-195	[130]	31	-	-	AUC 0.83
miR-335	[130]	31	-	-	AUC 0.87
Combi 6 miR’s ^5^	[136]	28	-	-	95% accuracy
miR-511 + miR-503	[111]	36	100	93	-
miR-511 + miR184	[111]	36	100	80	-
miR-483-3p + Smad4	[116]	50	-	92.8	-
Alterations 6 loci ^6^	[140]	138	100	83	-
>60% amplification chrom5	[137]	46	77.3	100	-
Combi >50 CNV’s + >10 LOH events	[137]	46	82	100	-

AUC = area under the curve, CNV = copy number variation, HU = Hounsfield units, miR = micro-RNA, MTO = metomidate, NLR = negative LR, NPV = negative predictive value, OR = odds ratio, PLR = positive LR, PPV = positive predictive value, LOH = loss of heterozygosity, LR = likelihood ratio, ^1^ Discriminative for adenomas, ^2^ For malignant disease (either ACC or adrenal metastasis) ^3^ For adrenal origin, ^4^ With a cut-off value of the Helsinki score of 8.5, ^5^ miR-503-5p, miR-483-3p, miR-450a-5p, miR-210, miR-483-5p, miR-421, ^6^ loci 5q, 7p, 11p, 13q, 16q, and 22q.

## Data Availability

Not applicable.

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
