# Peer review of "How to Differentiate Benign from Malignant Adrenocortical Tumors?"

_cancers, 2021, doi:10.3390/cancers13174383_

Round 1

Reviewer 1 Report

The authors have exemplified in figure 1 the most correct diagnostic procedure, corresponding to the current good clinical practice, in differentiating benign from malignant tumors, highlighting what is currently possible even only in reference centers and what is still under investigation.

  • It would be better to insert Figure 1 at the end of the text, after Table 4

Probably the greatest challenge is being able to distinguish benign from malignant adrenal masses before adrenalectomy in order to avoid unnecessary surgery

  • What is, according to the authors, the most specific, promising and easy tool for this goal?
  • Figure 2 is not very clear. From figure 2 it is not clear, as stated in the text, that the cut off of 20 HU is better than 10 HU in differentiating adenomas from carcinomas. What does N mean? What does it include?
  • Please check Table 2: in Helsinki score atypical mitosis inserted instead of mitosis, clear cell > 25% inserted instead of necrosis (check the numerical alignment).

Author Response

Reviewer 1

The authors have exemplified in figure 1 the most correct diagnostic procedure, corresponding to the current good clinical practice, in differentiating benign from malignant tumors, highlighting what is currently possible even only in reference centers and what is still under investigation.

It would be better to insert Figure 1 at the end of the text, after Table 4.

Figure 1 has been moved towards the conclusion section, where it has been placed right before table 4, instead of being mentioned already in the introduction section.

Probably the greatest challenge is being able to distinguish benign from malignant adrenal masses before adrenalectomy in order to avoid unnecessary surgery

What is, according to the authors, the most specific, promising and easy tool for this goal?

We agree that this is currently one of the greatest challenges for clinicians. In the conclusion section, we briefly summarize some of the most promising tools that are currently under research. We added some extra explanatory sentences in order to emphasize why we believe that radiological and urine parameters are very promising tools, as these tools could be used before surgery, and therefore could aid the decision on whether to perform adrenalectomy.

On page 15:

These developments are especially very promising as radiological parameters and urine metabolomics can be assessed preoperatively. Therefore these tools might aid clinicians in the decision on adrenalectomy.’

Figure 2 is not very clear. From figure 2 it is not clear, as stated in the text, that the cut off of 20 HU is better than 10 HU in differentiating adenomas from carcinomas. What does N mean? What does it include?

To address this issue, we expanded the caption of this figure (which is currently not figure 1 anymore because of the changed order of figures). In the caption we included a more detailed explanation of panel 2E and 2F, including of which tumor types N comprises. Moreover, a list of tumor types included as “other malignant tumors” and “other benign tumors” has been provided. Furthermore, we provided the number of false positive ACAs when using both HU cutoff values, thereby demonstrating why the cutoff value of HU>20 is better than HU>10 for differentiating ACC from ACA.

Legend figure 1:

‘The total number of patients N comprises of all patients with ACC, other malignant tumors, ACA and other benign tumors, of which the number of ACC patients is depicted in red. Of the group of adenomas, HU tumor attenuation was measured in 1328 patients. When using a cutoff HU≥10 for determining malignancy, a false positive result was found in 423 patients (31.9%). A cutoff HU≥20 yielded 200 false positive ACA patients (15.1%). Increasing the cutoff value from HU≥10 to HU≥20 leads to an increase in specificity for diagnosing ACC from 64% to 80%, therefore a cutoff HU of 20 is the most appropriate threshold for malignancy of adrenal tumors. These specific numbers cannot be extracted from the figure, since the total amount of patients also includes other entities of adrenal lesions.’

Please check Table 2: in Helsinki score atypical mitosis inserted instead of mitosis, clear cell > 25% inserted instead of necrosis (check the numerical alignment).

We increased the width of the table columns to ensure correct alignment of the table content.

Reviewer 2 Report

This manuscript provides a comprehensive review about traditional and experimental tools for differentiating adrenocortical adenoma (ACA) from adrenocortical carcinoma (ACC). The authors provide broad overview about tools and biomarkers, including clinically available and widely used methods such as imaging (dedicated CT scans, MRI, and PET-scan), histology (scoring systems and IHC), steroid biomarkers, and a myriad of experimental methods - most of them focused on molecular biomarkers. The text is well organized, and the authors provide a list of up-to-date references, and highlight the most important discussion points. 

Author Response

Reviewer 2

This manuscript provides a comprehensive review about traditional and experimental tools for differentiating adrenocortical adenoma (ACA) from adrenocortical carcinoma (ACC). The authors provide broad overview about tools and biomarkers, including clinically available and widely used methods such as imaging (dedicated CT scans, MRI, and PET-scan), histology (scoring systems and IHC), steroid biomarkers, and a myriad of experimental methods - most of them focused on molecular biomarkers. The text is well organized, and the authors provide a list of up-to-date references, and highlight the most important discussion points. 

Thank you for your positive comment.

Reviewer 3 Report

This manuscript summarizes the current diagnostic strategies and challenges in order to distinguish benign from malignant adrenal lesions. The manuscript is well-written and brings a topic of great importance, but some revisions should be performed.

Page 3 Line 93- The information is repeated, you should only use the abbreviation FDG-PET/CT.

Page 3 Lines 94-95 – The authors should justify how a chest ct can guide the clinical decision.

Figure 2 – The authors should mention the type of tumors that are included in the group defined as “other benign tumors”.

Figure 2 – Do the authors have the information regarding the number of adenomas, to include in the figures 2E and 2F? It would be interesting to see the number of adenomas currently identified using the two HU cut-offs.

Page 4 Lines 120-132 – Are there any cut of value for washout rates recommended by previous studies?

Page 5 line 142 - The authors should clarify what additional information can MRI bring for ACC diagnosis comparing to CT.

Page 6 line 197 - The sentence is incomplete “Additionally,…”.

Author Response

Reviewer 3

This manuscript summarizes the current diagnostic strategies and challenges in order to distinguish benign from malignant adrenal lesions. The manuscript is well-written and brings a topic of great importance, but some revisions should be performed.

We would like to thank reviewer 3 for the positive appraisal of the manuscript and addressed the raised issues in a point-to-point manner below.

Page 3 Line 93- The information is repeated, you should only use the abbreviation FDG-PET/CT.

We only used the abbreviation FDG-PET/CT instead of the full information.

Page 3 Lines 94-95 – The authors should justify how a chest ct can guide the clinical decision.

We agree that this is not completely clear from the text. The reason for performing a chest CT in case of suspicion of malignancy, namely to check the presence of metastatic disease, is now described in the manuscript.

Page 3:

‘In case of high suspicion of an ACC, a chest CT should be performed as well to assess the presence of any pulmonary metastases, because it can guide clinical decisions [11].

Figure 2 – The authors should mention the type of tumors that are included in the group defined as “other benign tumors”.

See explanation above.

We expanded the caption of this figure (which is nog figure 1 because of the changed order of figures). In the caption we included a more detailed explanation of panel 2E and 2F, including of which tumor types N comprises. Moreover, a list of tumor types included as “other malignant tumors” and “other benign tumors” has been provided. Furthermore, we provided the number of false positive ACAs when using both HU cutoff values, thereby demonstrating why the cutoff value of HU>20 is better than HU>10 for differentiating ACC from ACA.

Figure 2 – Do the authors have the information regarding the number of adenomas, to include in the figures 2E and 2F? It would be interesting to see the number of adenomas currently identified using the two HU cut-offs.

The number of adenomas with HU>10 and HU>20 is known, and is therefore added in the caption of the figure. However, we do not have information on the number of adenomas for each combination of tumor size and HU attenuation. Therefore, we did not add the number of ACA patients in the figure panels 2E and 2F itselves, but only in the caption.

Page 4 Lines 120-132 – Are there any cut of value for washout rates recommended by previous studies?

The commonly used cut-off values for washout rates have been added to the manuscript.

Page 4:

Cut-off values of an absolute washout >60% and a relative washout >40% have been proposed to accurately diagnose adenomas [19].

Page 5 line 142 - The authors should clarify what additional information can MRI bring for ACC diagnosis comparing to CT.

MRI is particularly useful to distinguish a small group of adenomas with minimal lipid content and therefore a HU value >10. These tumors can be assessed with chemical shift sequences on MRI. Besides, MRI is a sensitive tool to assess precise localization and separation from surrounding structures. This information has been added to the manuscript.

Page 4:

MRI can be used in particular for precise localization of the tumor and for assessment of separation from the surrounding structures. Furthermore, a small group of adenomas with minimal lipid content and therefore a HU value >10 can be distinguished with chemical shift sequences on MRI [23].

Page 6 line 197 - The sentence is incomplete “Additionally,…”.

We thank the reviewer for noticing this error. We incorporated the sentence that should be in the text after additionally, namely the fact that hormonal evaluation is always done in case of suspected ACC.

Page 5:

‘Additionally, for all patients with suspected ACC, a thorough and complete hormonal evaluation is recommended before surgery, regardless of the clinical phenotype [39].